# Comprehensive Analysis of 1-Year-Old Female Apolipoprotein E-Deficient Mice Reveals Advanced Atherosclerosis with Vulnerable Plaque Characteristics

**DOI:** 10.3390/ijms25021355

**Published:** 2024-01-22

**Authors:** Sotirios Kotsovilis, Maria Salagianni, Aimilia Varela, Constantinos H. Davos, Ioanna E. Galani, Evangelos Andreakos

**Affiliations:** 1Laboratory of Immunobiology, Center for Clinical Research, Experimental Surgery and Translational Research, Biomedical Research Foundation, Academy of Athens, GR 11527 Athens, Greece; skotsovilis@bioacademy.gr (S.K.); msalagianni@bioacademy.gr (M.S.); igalani@bioacademy.gr (I.E.G.); 2Cardiovascular Research Laboratory, Center for Clinical Research, Experimental Surgery and Translational Research, Biomedical Research Foundation, Academy of Athens, GR 11527 Athens, Greece; evarela@bioacademy.gr (A.V.); cdavos@bioacademy.gr (C.H.D.)

**Keywords:** animal models, atherosclerosis, inflammation, macrophages, apolipoprotein E-deficient mice, cardiovascular disease

## Abstract

Apolipoprotein E-knockout (*Apoe*-/-) mice constitute the most widely employed animal model of atherosclerosis. Deletion of *Apoe* induces profound hypercholesterolemia and promotes the development of atherosclerosis. However, despite its widespread use, the *Apoe*-/- mouse model remains incompletely characterized, especially at late time points and advanced disease stages. Thus, it is unclear how late atherosclerotic plaques compare to earlier ones in terms of lipid deposition, calcification, macrophage accumulation, smooth muscle cell presence, or plaque necrosis. Additionally, it is unknown how cardiac function and hemodynamic parameters are affected at late disease stages. Here, we used a comprehensive analysis based on histology, fluorescence microscopy, and Doppler ultrasonography to show that in normal chow diet-fed *Apoe*-/- mice, atherosclerotic lesions at the level of the aortic valve evolve from a more cellular macrophage-rich phenotype at 26 weeks to an acellular, lipid-rich, and more necrotic phenotype at 52 weeks of age, also marked by enhanced lipid deposition and calcification. Coronary artery atherosclerotic lesions are sparse at 26 weeks but ubiquitous and extensive at 52 weeks; yet, left ventricular function was not significantly affected. These findings demonstrate that atherosclerosis in *Apoe*-/- mice is a highly dynamic process, with atherosclerotic plaques evolving over time. At late disease stages, histopathological characteristics of increased plaque vulnerability predominate in combination with frequent and extensive coronary artery lesions, which nevertheless may not necessarily result in impaired cardiac function.

## 1. Introduction

Atherosclerosis is a chronic disease that develops in the walls of large- and medium-sized arteries throughout the cardiovascular system and is initiated and aggravated by hypercholesterolemia and inflammation [1,2,3,4,5,6]. Hypercholesterolemia followed by subtle alterations in the arterial wall environment, triggered by lipid deposition, inflammatory cell influx, and secretion of inflammatory mediators, are currently established as etiological factors of atherosclerosis [4,7,8]. Endocrine factors and related disorders have also been described as being involved in the development of atherosclerosis both systemically and vascularly, such as hypothyroidism, diabetes, and depression [9]. Atherosclerosis is the leading cause of cardiovascular disease and the main cause of disability and premature mortality globally, accounting for over one third of all global deaths annually [10].

During the past three decades, the concept of the inflammatory etiology of atherosclerosis [3] has prevailed over older ideas such as the response-to-injury hypothesis or the degenerative disease hypothesis [11,12]. A valuable contribution to this was made by the introduction of novel experimental methods such as gene targeting technology [13,14], which enabled the development of mouse models of atherosclerosis, namely the apolipoprotein E-knockout mouse model (*Apoe*-/-) [15,16,17] and the low-density lipoprotein (LDL) receptor-knockout mouse model (LDLR) [18]. In a recent study, LDLR-/- mice crossbred with a fibrillin-1 heterozygous mutation via homologous recombination were used to study the pathological and physiological mechanisms of advanced atherosclerotic unstable plaques [19]. The creation of these models commenced a new era in atherosclerosis research, as wild-type C57BL/6 mice fed either normal or high-fat diets are rather resistant to atherosclerosis due to the protective role of high-density lipoprotein (HDL) in this process, hampering the unwinding of the mechanisms involved in driving disease. However, the deletion of the *Apoe* gene in knockout mice, even under normal chow diets, results in an enhanced accumulation of liver-derived very-low-density lipoprotein (VLDL) remnants, enhanced plasma cholesterol levels, and the spontaneous development of aortic atherosclerotic plaques analogous to those observed in human subjects, mainly with respect to lesion progression and cell types [17,20,21,22,23,24]. Nonetheless, although the *Apoe*-/- mouse model has been employed extensively to address various specific objectives pursued in a plethora of studies on atherosclerosis, detailed studies on the characterization of the model itself, especially at late timepoints, have been more limited [17,21,22,23,24,25]. Currently, there is one study that has employed aged (1-year-old) *Apoe*-/- mice to determine the contribution of fibronectin containing an extra domain A to plaque destabilization in the innominate artery [26].

A number of studies have previously reported serum biochemical data [27,28,29], macrophage and smooth muscle cell presence [29,30], calcification [29,30] or necrotic core formation within atherosclerotic plaques [29,31,32], as well as data on cardiac or arterial function derived from Doppler ultrasonography [29,33,34,35,36,37]. These, however, were restricted to specific snapshots of the disease process and did not provide information about the longitudinal progression of atherosclerosis from the early stages up to, and including, the late disease stages. Thus, there are very few studies in that respect, and even then, important information and specific readouts are not reported [21,22,23]. This includes serum biochemical parameters, quantitative data on the proportions of cell types, the frequency of calcification, the area of necrotic cores within atherosclerotic plaques, or the partial or total occlusion of coronary arteries throughout the entire period extending from the early stages to the late stages of atherosclerosis. In cases of studies assessing some of these characteristics [21,22,23], other limitations are present, such as the absence of quantification of parameters using adequately large sample sizes of *Apoe*-/- mice. Moreover, a histomorphometric analysis of coronary arteries in the *Apoe*-/- mouse model has not been reported.

Therefore, the current study was designed to address long-standing deficiencies in the literature of the past three decades and provide missing information through the characterization of atherosclerosis and the evaluation of cardiac and carotid artery function in the *Apoe*-/- mouse model. A comprehensive analysis was performed by selecting time points for examination (10, 26, and 52 weeks of age of *Apoe*-/- mice) that were anticipated to thoroughly represent and recapitulate the progression of atherosclerosis from the early to the late stages of the disease process. Particular attention was given to late disease stages, where major histopathological changes linked to the transition of “stable” subclinical atherosclerotic lesions into “vulnerable” clinically potentially hazardous lesions were characterized. Importantly, histomorphometric analysis of atherosclerotic lesions in coronary arteries was also performed by developing a novel method. The accumulation of this basic information was deemed important both from a research and clinical standpoint.

## 2. Results

### 2.1. Atherosclerosis in Apoe-/- Mice Is a Highly Dynamic Process in Which Atherosclerotic Plaques Evolve over Time

The time-course of development and progression of atherosclerosis was studied in *Apoe*-/- mice at time points representing early (10 weeks of age), intermediate (26 weeks of age), and late (52 weeks of age) stages of the disease (Figure 1a–e). Statistical analysis using the unpaired Student’s *t*-test revealed that the body weight of mice was statistically significantly increased at either 26 or 52 weeks compared with body weight at 10 weeks (Figure 1a). This was not the case in total cholesterol serum concentration, which did not change in different ages of *Apoe*-/- mice (Figure 1b). Furthermore, the VLDL, LDL, and HDL levels were not statistically different amongst the groups (Appendix A). Using the same statistical test, the serum concentration of triglycerides was statistically significantly lower at 26 weeks (*p* = 0.003 < 0.01) compared with that at 10 weeks and with no statistical significance compared with that at 52 weeks. Serum triglyceride levels were also lower in wild-type (WT) mice at 26 weeks of age compared to *Apoe*-/- mice at 10, 26, and 52 weeks of age (Figure 1c). As *Apoe*-/- mice aged, progressively larger atherosclerotic lesions developed, as revealed by Oil Red O (ORO)-staining of serial cross sections at the level of the aortic root (Figure 1d) and subsequent quantification of atherosclerotic lesion area/lesion size using morphometric analysis of stained sections (Figure 1e). Using the unpaired Student’s *t*-test, lesion area was statistically significantly increased at 26 weeks than at 10 weeks (*p* = 0.001), and additionally at 52 weeks than at 26 weeks (*p* = 0.0008; Figure 1e). As revealed by one-way analysis of variance, with respect to body weight (Figure 1a), triglycerides (Figure 1c), and lesion area (Figure 1e), differences among 10-, 26-, and 52-week-old mice were statistically significant (*p* < 0.0001, *p* = 0.0173, and *p* < 0.0001, respectively). With respect to total cholesterol (Figure 1b), differences among 10-, 26-, and 52-week-old mice did not reach statistical significance (*p* = 0.1303). As expected, sections from WT animals were negative for ORO staining (Figure 1d). These findings indicate that atherosclerosis in *Apoe*-/- mice fed a normal chow diet is a highly dynamic process in which atherosclerotic plaques evolve and progressively enlarge over time. The progression of atherosclerosis towards late disease stages is characterized by elevated lipid deposition in atherosclerotic plaques, which does not seem to be associated with an analogous increase in serum lipid concentrations with the advent of time (Figure 1a–e).

### 2.2. Atherosclerotic Lesions of Apoe-/- Mice Acquire Morphological Characteristics of a More “Vulnerable Phenotype” at Late Disease Stages

Although the lesion area accurately reflects the progression of atherosclerosis, it is the morphology of atherosclerotic plaques that is generally considered to be a more important marker and predictor of atherosclerotic plaque instability, vulnerability, or disruption, and acute clinical events in patients, such as acute myocardial infarction [38,39,40]. Therefore, we evaluated some major morphological characteristics during the progression of atherosclerosis as surrogate markers for lesion vulnerability (Figure 2 and Figure 3). Aortic valve atherosclerotic lesions in 52-week-old *Apoe*-/- mice fed a normal chow diet exhibited a reduced percentage of lipid deposition out of the atherosclerotic lesion area compared with 26-week-old *Apoe*-/- mice, as revealed by ORO staining of serial cross sections at the level of the aortic root and subsequent morphometric analysis/quantification of the atherosclerotic lesion area (Figure 2a,b). Statistical analysis using the unpaired Student’s *t*-test revealed that this reduction did not reach statistically significant levels, though there was a tendency towards this effect (*p* = 0.075; Figure 2b). In agreement with this histological finding, immunofluorescence showed that the percentage of CD68-positive macrophages (Figure 2c) as well as the percentage of alpha smooth muscle actin-positive cells out of the atherosclerotic lesion area (Figure 2d) were both statistically significantly decreased in 52-week-old *Apoe*-/- mice compared with 26-week-old *Apoe*-/- mice (*p* = 0.001 and *p* = 0.01, respectively). It is noteworthy that 52-week-old *Apoe*-/- mice exhibited increased necrotic core area/size—defined as acellular 4′,6-diamidino-2-phenylindole (DAPI)-negative areas containing remnants of cells and extracellular lipid—compared to that of 26-week-old *Apoe*-/- mice (Figure 3a–d). This was highly significant (*p* < 0.0001; Figure 3c). However, the percentage of necrotic core area out of atherosclerotic lesion area was not significantly higher in 52-week-old mice than in 26-week-old *Apoe*-/- mice (*p* = 0.225; Figure 3d). These findings indicate that in *Apoe*-/- mice fed a normal chow diet, the progression of atherosclerosis towards late disease stages is coincident with the transition of atherosclerotic lesions of the aortic valve from a more cellular macrophage-rich and less necrotic phenotype at 26 weeks of age to a less cellular and more necrotic phenotype at 52 weeks of age (Figure 2a–c and Figure 3a–d). Thus, atherosclerotic lesions acquire morphological characteristics attributed to a more “vulnerable phenotype” [38,41], which is more prone to disruption and acute clinical episodes.

### 2.3. Aged Apoe-/- Mice Develop Plaque Calcification

Calcified nodules have been identified in 2–7% of thrombi in cases of sudden death in human populations [5]. Calcification in the aortic root has been observed as early as 16 weeks [42] or at the intermediate time points of 20 weeks [43] of age in *Apoe*-/- mice fed a normal chow diet. However, the frequency of this early or intermediate calcification development has never been reported, and, furthermore, one study has detected aortic root calcification in *Apoe*-/- mice fed a normal chow diet only at late disease stages, such as 60 weeks [25]. We therefore studied the development of calcification in atherosclerotic plaques of the aortic valve in 26- and 52-week-old *Apoe*-/- mice using Alizarin Red S-stained (Figure 4a) and ORO-stained (Figure 4b) serial cross sections from the aortic root and subsequently morphometric analysis of stained sections (Figure 4c). We found that at 52 weeks, 4 out of 10 *Apoe*-/- mice (40%; Figure 4c) exhibited calcium deposits in the intima of atherosclerotic plaques, whereas at 26 weeks, only 1 out of 10 mice (10%; Figure 4c) presented with such calcified areas. The presence of calcification within coronary arteries was not observed on the same stained sections (Appendix A). These findings indicate that in *Apoe*-/- mice fed a normal chow diet, the progression of atherosclerosis towards late disease stages is coincident with a substantial increase in the frequency of the presence, but not necessarily also of the area, of calcification in atherosclerotic lesions of the aortic valve (Figure 4a–c).

### 2.4. Aged Apoe-/- Mice Develop Coronary Occlusion

Since many decades ago, it has been well documented that a causal factor in acute clinical cardiovascular events, such as acute myocardial infarction, is the partial or total occlusion of the lumen of coronary arteries. Still, this is rarely addressed in *Apoe*-/- mouse models of atherosclerosis. We thus identified and assessed coronary arteries and their branches in ORO-stained serial cross sections prepared at the level of the aortic root (Figure 5a–c). The right coronary artery (depicted as artery 1 in Figure 5a) is distant from the remaining arteries on the contralateral side of each light photomicrograph, which represent branches of the left coronary artery (arteries 2–4 in Figure 5a). These include the left anterior descending coronary artery (LADCA; artery 2 in Figure 5a), the circumflex coronary artery (CIRCUMFLEX; artery 3 in Figure 5a), and the left main coronary artery (LMCA; artery 4 in Figure 5a).

Morphometric analysis of these sections revealed that the lesion area (Figure 5d), percentage of lesion area out of the total cross-sectional arterial area (Figure 5e), and percentage of lipid deposition out of the lesion area (Figure 5f) were all substantially higher for all three left coronary artery major branches (LADCA, CIRCUMFLEX, and LMCA) in 52-week-old *Apoe*-/- mice than in 26-week-old *Apoe*-/- mice. A considerable proportion of the lumen of all left coronary artery major branches was encountered to be occluded by atherosclerotic plaques in 52-week-old *Apoe*-/- mice (median values ranging from 18.84% for LMCA to 49.07% for LADCA; Figure 5e), in contrast to 26-week-old *Apoe*-/- mice, in which atherosclerotic plaques in all left coronary artery major branches most frequently were absent or rarely were present, but had a minimal extent (Figure 5e).

Taken together, these findings suggest that in *Apoe*-/- mice fed a normal chow diet, the progression of atherosclerosis towards late disease stages is coincident with the ubiquitous development of extensive atherosclerotic lesions in all three left coronary artery major branches (LADCA, CIRCUMFLEX, and LMCA) and that the resultant partial occlusion of the lumen of coronary arteries by atherosclerotic plaques is a histopathological characteristic of late disease stages, but not intermediate or early stages (Figure 5a–f).

### 2.5. One-Year Old Apoe-/- Mice Exhibit Higher Carotid Atherosclerosis, but Their Cardiac Function Is Not Affected

The evaluation of cardiac function in 26-week-old and 52-week-old *Apoe*-/- mice fed a normal chow diet compared with C57BL/6J wild-type control mice of the same age, gender, and diet was performed in order to address the issue of whether and to what extent the histopathological changes that occur concurrently with the progression of atherosclerosis can affect cardiac function. Doppler ultrasound analysis and subsequent pairwise comparisons of murine groups using the unpaired Student’s *t*-test revealed that heart rate (*p* = 0.003 < 0.01), posterior wall thickness (*p* = 0.009 < 0.01), and mean carotid velocity (*p* = 0.027 < 0.05) were statistically significantly higher, while carotid pulsatility index was statistically significantly lower (*p* = 0.019 < 0.05) in 52-week-old *Apoe*-/- mice compared with 26-week-old *Apoe*-/- mice (Table 1). The interpretation of these differences, however, is not clear, given that the same parameters were not statistically significantly different (*p* > 0.05) between 52-week-old *Apoe*-/- mice and 52-week-old C57BL/6 wild-type control mice (Table 1). Thus, the overall left ventricular function did not appear to differ in 52-week-old *Apoe*-/- mice from that in 52-week-old C57BL/6 wild-type control mice (Table 1). Taken together, these findings indicate that in *Apoe*-/- mice fed a normal chow diet, the extensive progression of atherosclerosis at late timepoints may not necessarily lead to impaired cardiac function (Table 1).

## 3. Discussion

The present study addresses long-standing deficiencies in the literature of the past three decades and provides missing information by characterization of atherosclerosis and evaluation of cardiac and carotid artery function in the most widely used animal model of atherosclerosis, the *Apoe*-/- mouse model. A comprehensive analysis was performed by selecting time points for investigation (10, 26, and 52 weeks of age of *Apoe*-/- mice) that were considered to thoroughly represent and recapitulate the progression of atherosclerosis from the early stages up to and including late disease stages. In comparison to previous studies addressing this issue [21,22,23], this study reports for the first time an extensive amount of significant information concerning various characteristics of the *Apoe*-/- mouse model, such as serum biochemical parameters, quantitative data on the proportions of cell types (macrophages and smooth muscle cells), frequency of calcification, and area of necrotic cores within atherosclerotic plaques, level of occlusion of the coronary arteries, and Doppler ultrasonographic data on cardiac or carotid artery function. To the best of our knowledge, this is the first histomorphometric analysis of atherosclerotic lesions in coronary arteries in the *Apoe*-/- mouse model.

Results revealed that atherosclerosis in *Apoe*-/- mice is a highly dynamic process in which atherosclerotic plaques evolve and progressively enlarge over time. These findings are consistent with preceding studies in the literature, describing the main successive histopathological stages during atherosclerotic lesion development [6,20,23,44,45]. Specifically, we found that 10-week-old *Apoe*-/- mice presented with atherosclerotic plaques at the level of the aortic valve that exhibited histopathological characteristics such as moderate macrophage presence, intracellular lipid accumulation, and foam cell formation similar to those described for fatty streaks or type II atherosclerotic lesions in humans according to the American Heart Association histological classification of atherosclerosis [45]. Furthermore, we found that 26-week-old *Apoe*-/- mice exhibited aortic valve atherosclerotic plaques bearing histopathological characteristics similar to those described for preatheromata/intermediate/transitional/type III lesions or for “advanced” atherosclerotic lesions in humans, comprising either atheromata/type IV lesions or fibrous/type V lesions [44,45]. These included large or multiple lipid cores with both intracellular and extracellular lipid accumulation and a prominent macrophage presence, especially in the shoulder and cap areas of the plaque. Finally, 52-week-old *Apoe*-/- mice exhibited aortic valve plaques with histopathological characteristics similar to those described for “advanced” atherosclerotic lesions in humans, namely fibrous/type V lesions [44]. Such lesions have been reported to have multiple lipid cores with limited macrophage presence and marked extracellular lipid deposition, as well as large acellular necrotic areas and calcium deposits, and additionally possess the potential to develop fissures, hematomata, and/or thrombi (type VI lesions), thus causing clinical manifestations [44]. Importantly, only lesions in the last category can have clinical relevance and implications, whereas preceding lesions characterize preclinical stages of atherosclerosis [44].

In our study, the progression of atherosclerosis towards late disease stages is coincident with the transition of atherosclerotic lesions of the aortic valve from a more cellular macrophage-rich and less necrotic phenotype at 26 weeks of age to a less cellular and more necrotic phenotype at 52 weeks of age. Whether this is also linked to higher inflammation in the plaque of 52-week-old mice is not clear at this stage. Necrotic material and lipid deposition can certainly drive the activation of innate inflammatory responses in the plaque through the stimulation of pattern recognition receptors and the production of pro-inflammatory cytokines, whereas reduced macrophage presence or ‘alternative’ macrophage activation to M2-like phenotypes can limit this process [29,46,47]. Further studies are therefore needed in that respect.

Our findings corroborate previous results reported by our laboratory for male 26-week-old *Apoe*-/- mice [29] and expand the investigation up to the late disease stage of 52 weeks. Thus, atherosclerotic lesions acquire morphological characteristics attributed to a more “vulnerable phenotype” [38,39,40,41]; this is the predominant clinicopathological phenomenon by which “stable” subclinical atherosclerotic lesions undergo a gradual or sudden transformation into more unstable, prone to disruption and acute clinical episodes, “vulnerable” and clinically potentially hazardous lesions [48]. Hence, this conversion is regarded as a “critical turning point” in the progression of atherosclerosis [48].

Another interesting parameter the present study addresses is aortic valve calcification. This was detected in only 10% of the population of 26-week-old *Apoe*-/- mice, suggesting that only a limited minority of lesions were of the Vβ type [44], while 40% of the population of 52-week-old *Apoe*-/- mice presented with aortic valve calcification, indicating that a considerable proportion of lesions were of the Vβ type [44]. Hence, the majority (60%) of the murine population presented with advanced atherosclerotic lesions in the aortic valve without calcification. Therefore, it appears that the presence of aortic valve calcification, in spite of its considerable frequency, does not necessarily histopathologically characterize late disease stages. In a previous study, aortic sinus calcification was present in 85.71% (6/7) of 41- to 42-week-old *Apoe*-/- mice and 58.33% (7/12) of male *Apoe*-/- mice, both fed a normal chow diet [49].A possible explanation for this variability in calcification frequency might be the variability in atherosclerotic lesion area observed in the latter study [49], which was also observed in our study.

The present study additionally demonstrated that in *Apoe*-/- mice, the progression of atherosclerosis towards late disease stages is coincident with the ubiquitous development of extensive atherosclerotic lesions in all three left coronary artery major branches (LADCA, CIRCUMFLEX, and LMCA) and that the ensuing partial occlusion of the lumen of coronary arteries by atherosclerotic plaques is a histopathological characteristic of late disease stages but not intermediate or early stages. These findings are in agreement with a previously mentioned theory, according to which, at late or ultimate stages of atherosclerosis, atherosclerotic plaques might become unstable and rupture, exposing subendothelial prothrombogenic material to platelets of blood circulation, thereby provoking blood clots, thrombus formation, and arterial luminal occlusion [5].

The characteristics of the “vulnerable phenotype” of atherosclerotic lesions described in this report were surrogate endpoints, thus being substitutes for true clinical endpoints. It has been pointed out that atherosclerotic plaques can be described as “vulnerable”, even in the absence of plaque rupture, if characteristics of vulnerability are present [40]. It might also be noted that the sequence of events that characterize the transition from atherosclerosis to acute clinical episodes has already been described by previous enlightening studies both in murine models and in humans [6,38,40,41,50].

A limitation of our study is that it was performed on female mice. However, although sexual dimorphism is an important parameter of cardiovascular disease in humans, in *Apoe*-deficient mice, this is more obscure. Although the male sex has been shown to be a risk factor for more severe cardiovascular damage in some studies [51,52,53], in others, this made no difference [54,55] or even appeared to be protective [56]. With respect to lesion size, we did not find differences between male and female mice ourselves (Appendix A). Nevertheless, similar studies in male mice are needed to address whether these observations also stand there.

In summary, our study expands the spectrum of current knowledge on the *Apoe*-/- mouse model. We mostly centered on late disease stages and described some major histopathological changes characterizing the transition of “stable” subclinical atherosclerotic lesions into “vulnerable” clinically potentially hazardous lesions. From a clinical perspective, it appears important that preventive plaque-stabilizing strategies directed against the progression of atherosclerotic lesions towards vulnerability might prove in the future to be quite as promising, if not more efficacious, than preventive strategies against the initiation of atherosclerosis per se [32]. Finally, from a research perspective, the present study might offer useful basic information for future research on the *Apoe*-/- mouse model, considering that this research issue is far from being exhausted. For example, future studies may examine atherosclerotic lesions in coronary arteries only at late disease stages, given that in this study no coronary lesions were identified at intermediate stages, such as the time point of 26 weeks of age. This study might also have implications in studying mechanisms of atherosclerosis progression and plaque vulnerability, as well as in the development and testing of novel compounds in the pharmaceutical industry; for example, drugs inhibiting coronary atherosclerosis presenting with vulnerable plaque characteristics, such as calcification or necrotic core formation.

## 4. Materials and Methods

### 4.1. Animals

Experimental animals in this study comprised 10-, 26-, and 52-week-old female *Apoe*-/- mice on a C57BL/6J background. Μice were obtained from The Jackson Laboratory (Bar Harbor, Maine, Sacramento, CA, USA) and were bred in-house. Mice were fed a normal chow diet containing 18.6% protein and 6.2% fat (2018 Teklad Global 18% Protein Rodent Diet, Harlan Laboratories, Inc., Indianapolis, IN, USA) as previously described [29]. Mice were housed in individually ventilated cages under specific pathogen-free conditions at the Animal House Facility of the Biomedical Research Foundation of the Academy of Athens. This study was approved by the Bioethics Committee for the Use of Experimental Animals of the Biomedical Research Foundation of the Academy of Athens and is reported in accordance with ARRIVE guidelines (https://arriveguidelines.org).

### 4.2. Histological Analysis

Following blood collection, the heart of each mouse was perfused with phosphate-buffered saline (PBS) and collected. For histological analysis of the aortic valve area, hearts were embedded in an optimal cutting temperature compound after overnight fixation in 4% paraformaldehyde in PBS and equilibration in 30% sucrose in distilled water for a further time period of 24 h. Cryostat 10 μm thick serial sections of the aortic valve were prepared, spanning a 600-μm area. Stainings using ORO (Sigma-Aldrich Corporation, St. Louis, MO, USA) to study lipid deposition throughout the aortic valve as well as 2% Alizarin Red S solution (Sigma-Aldrich Corporation, St. Louis, MO, USA) to study calcification throughout the aortic valve were performed on cryostat sections. In both ORO and Alizarin Red S stainings, counterstaining with hematoxylin (HX) was performed. The histological sections were observed with a DMLS2 optical microscope equipped with a DFC500 camera (Leica Microsystems GmbH, Wetzlar, Germany).

### 4.3. Immunohistochemistry and Immunofluorescence Staining

Aortic valve cryosections were incubated with a rat anti-mouse CD68 monoclonal antibody (clone FA-11; AbDSerotec, Bio-Rad Laboratories, Inc., Hercules, CA, USA) and a Cy3-conjugated anti-alpha smooth muscle actin monoclonal antibody (anti-αSMA; clone 1A4, Sigma-Aldrich Corporation, St. Louis, MO, USA). Control sections comprised an isotype control section incubated with rat IgG (Jackson ImmunoResearch Laboratories, Inc., West Grove, PA, USA), as well as an unstained control section. Immunofluorescence using anti-CD68 or anti-αSMA monoclonal antibodies is aimed at studying the infiltration of atherosclerotic plaques with macrophages or the accumulation of smooth muscle cells within plaques, respectively. Blocking of non-specific binding was achieved by incubation with 10% goat serum in PBS-0.1% polyethyleneglycol-mono-[p-(1,1,3,3-tetramethylbutyl)-phenyl]-ether/Triton X-100 (Triton^®^ X-100 BioChemica, AppliChem GmbH, Darmstadt, Germany; now integrated into PanReac AppliChem, ITW Reagents). Isotype controls were used at the same concentrations as the respective primary antibodies. Isotype control sections were incubated with a goat anti-rat Alexa 594-conjugated antibody (Molecular Probes, Invitrogen, Eugene, OR, USA) as a secondary antibody, and test sections were incubated with the same secondary antibody in combination with a Cy3-conjugated anti-αSMA monoclonal antibody (clone 1A4, Sigma-Aldrich Corporation, St. Louis, MO, USA). Unstained sections were incubated with 4% goat serum in PBS-0.1% polyethyleneglycol-mono-[p-(1,1,3,3-tetramethylbutyl)-phenyl]-ether/Triton X-100 (Triton^®^ X-100 BioChemica, AppliChem GmbH, Darmstadt, Germany; now integrated into PanReac AppliChem, ITW Reagents), in place of a secondary antibody. All isotype control sections, as well as unstained control sections, exhibited an absence of a positive signal (Appendix A). Nuclear counterstaining was performed with DAPI (Molecular Probes, Invitrogen, Eugene, OR, USA). Mounting was performed using a fluorescence mounting medium (Dako Fluorescent Mounting Medium, Dako North America, Inc., Carpinteria, CA, USA). Confocal images were acquired by employing a confocal microscope (Leica TCS SP5, Leica Microsystems, Bensheim, Germany).

### 4.4. Morphometry/Quantification of Aortic Lesion Area, Calcified Area, Macrophage Infiltration, Smooth Muscle Cell Accumulation, and Plaque Necrosis

For all cryosections stained with ORO or Alizarin Red S or incubated with anti-CD68 or anti-αSMA monoclonal antibodies, positively stained areas were quantified using a public domain image processing software program (ImageJ 1.37c, developed by Wayne Rasband, retired from the Research Services Branch, National Institute of Mental Health, Bethesda, MD, USA), as described previously [29]. Plaque necrosis was determined by drawing boundary lines around regions free of DAPI staining and quantifying the region area, which represented the necrotic core area, using the same software program, as described in our preceding study [29]. Region areas lower than a 3000-μm2 threshold were considered not to constitute substantial areas of necrosis and therefore were excluded from the analysis, as suggested by a preceding study [31].

### 4.5. Histology and Morphometry of Coronary Arteries

ORO staining of serial cross sections prepared at the level of the aortic root was performed in order to examine the morphological characteristics of atherosclerotic plaques in coronary arteries in female 26- and 52-week-old *Apoe*-/- mice. On light photomicrographs of these sections, coronary arteries and their branches were histologically discriminated and identified using a standardized procedure. The isolated artery, which appeared distant from the remaining arteries, was the right coronary artery. On the contralateral side of each light photomicrograph, the branches of the left coronary artery were located. These branches were the left anterior descending coronary artery (LADCA), the circumflex coronary artery (CIRCUMFLEX), and the left main coronary artery (LMCA), which formed as the convergence of LADCA and CIRCUMFLEX. The distinction between LADCA and CIRCUMFLEX was drawn on the basis of the fact that LADCA appeared closer than CIRCUMFLEX to the right coronary artery on light photomicrographs. Morphometric analysis of ORO-stained serial cross sections prepared at the level of the aortic root was based on the calculation of three parameters: lesion area, the percentage of lesion area out of the total cross-sectional arterial area, and the percentage of lipid deposition out of the lesion area. For each ORO-stained serial cross section prepared at the level of the aortic root, the lesion area was defined as the area (measured in μm2) occupied by the atherosclerotic plaque formed within each coronary artery branch. For each ORO-stained serial cross section prepared at the level of the aortic root, the percentage of lesion area out of the total cross-sectional arterial area was calculated by dividing the lesion area by the total cross-sectional arterial area and subsequently by multiplying by 100%. Finally, for each ORO-stained serial cross section prepared at the level of the aortic root, the percentage of lipid deposition out of the lesion area was calculated by dividing the area occupied by lipids by the lesion area and, subsequently, by multiplying by 100%.

### 4.6. Serum Biochemistry

For serum lipid concentration measurements, mice were fasted for 16 h. Subsequently, body weight was measured, and immediately blood was collected from the orbital sinuses of 10-, 26-, and 52-week-old *Apoe*-/- mice on a C57BL/6J background and centrifuged at 13,000 rpm for 20 min; separated serum samples were stored at −80 °C until further analysis. Serum concentrations of total cholesterol and triglycerides were determined using the commercially available Cholesterol or Triglyceride Assay Kit, respectively (Cayman Chemical Company, Ann Arbor, MI, USA), according to the manufacturer’s instructions. HDL-cholesterol levels as well as LDL/VLDL levels were measured according to the manufacturer’s protocol, based on an enzymatic colorimetric assay with a commercial kit from Abcam (ab65390; Cambridge, MA, Canada).

Doppler Ultrasonography. Echocardiographic studies were performed in mice anesthetized with intraperitoneal injection of 100 mg/kg ketamine using a Vivid 7 GE ultrasound system (GE Healthcare, Little Chalfont, UK) with a 13 MHz linear transducer and a 6 MHz pulsed Doppler probe in order to measure transvalvular aortic blood flow velocities. Two-dimensional targeted M-mode imaging was obtained from the short-axis view at the level of the greatest left ventricular dimension. Images were analyzed using the EchoPAC PC SW 3.1.3 software (GE Healthcare, Little Chalfont, UK). M-mode measurements of left ventricular end-diastolic diameter (EDD), left ventricular end-systolic diameter (ESD), and left ventricular posterior wall thickness at diastole (PWT) were made. End diastole was determined at the maximal left ventricular diastolic dimension, and end systole was defined at the peak of posterior wall motion. For each measurement, three beats were averaged. All ultrasonographic parameters mentioned in Table 1 were calculated as described in a previous study by our research team [29]. Finally, carotid artery Doppler flow waveforms were recorded, and the maximum, minimum, and mean velocities of the waveforms were measured; the pulsatility index was calculated as described in our preceding study [29].

### 4.7. Statistical Analysis

The normality of all statistical distributions was evaluated using the Kolmogorov–Smirnov test (with a Dallal–Wilkinson–Lilliefor *p* value), D’Agostino’s and Pearson’s omnibus K2 test, and the Shapiro–Wilk test. Statistical comparisons between study groups were performed using the unpaired Student’s *t*-test (in cases of normal distributions) or the Mann–Whitney U test (in cases of distributions that were not normal). A one-way analysis of variance was used to test for differences among at least three groups, since the two-group case was covered by a Student’s *t*-test. Differences between study groups were considered to be statistically significant whenever the *p* value was ≤0.05. All statistical tests were two-tailed and were performed using a commercially available statistical software program (GraphPad Prism, version 5.00 for Windows, GraphPad Software, Inc., San Diego, CA, USA). The investigators who performed all experiments, assessments, and measurements (A.V. for Doppler ultrasonography and S.K. for all other experiments, assessments, and measurements) were masked with respect to the results of the study; thus, both examiner and operator masking were present.

## Figures and Tables

**Figure 1 ijms-25-01355-f001:**
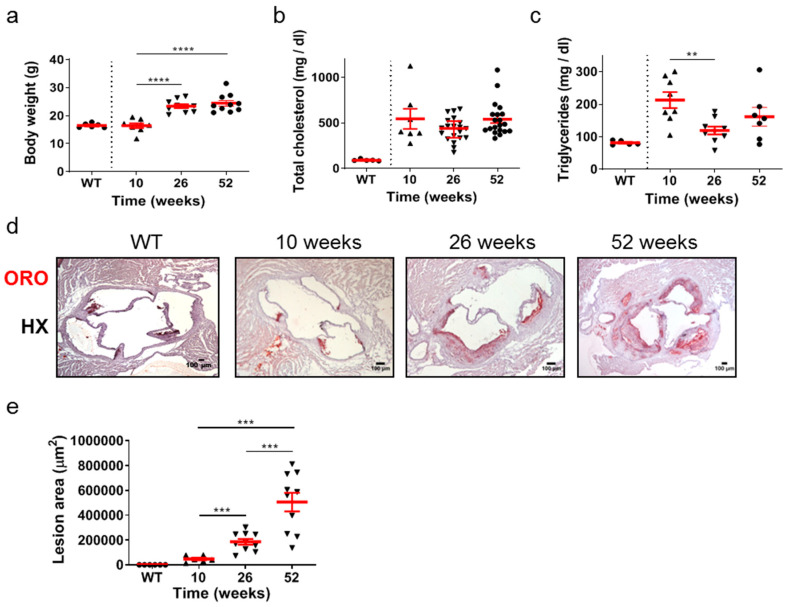
Atherosclerosis in *Apoe*-/- mice fed a normal chow diet is a highly dynamic process in which atherosclerotic plaques evolve over time. The progression of atherosclerosis towards late disease stages is characterized by enhanced lipid deposition in atherosclerotic plaques, which does not seem to be associated with an analogous increase in serum lipid concentrations with the advent of time. (**a**) Body weight of 10- (triangles), 26- (triangles), and 52-week-old (circles) *Apoe*-/- mice (*n* = 8, 10, and 10 mice for each group, respectively) and wild-type C57BL/6 mice (*n* = 6, 10 weeks old, circles). (**b**,**c**) Serum concentrations of total cholesterol and triglycerides in 10- (triangles), 26- (triangles), and 52-week-old (circles) *Apoe*-/- mice (in (**b**), *n* = 7, 20, 20 mice; in (**c**), *n* = 8, 9, 7 mice for each group, respectively) and wild-type C57BL/6 mice (*n* = 6, 26 weeks old, circles). (**d**) Representative light photomicrographs of ORO-stained cross sections from the aortic root of 10-, 26-, and 52-week-old *Apoe*-/- mice, demonstrating lipid deposition at the level of all three aortic valve leaflets. HX indicates hematoxylin. (**e**) Atherosclerotic lesion area determined using morphometric analysis of ORO-stained serial cross sections from the aortic root of 10- (triangles), 26- (triangles), and 52-week-old (triangles) *Apoe*-/- mice by ImageJ software (ImageJ 1.37c, developed by Wayne Rasband, retired from the Research Services Branch, National Institute of Mental Health, Bethesda, MD, USA) (*n* = 6, 10, and 10 mice for each group, respectively). In (**a**,**c**,**e**), mean value ± standard error of the mean (SEM) is displayed in red. In (**b**), mean value ± SEM is denoted in red. Statistical comparisons between groups were performed using the unpaired Student’s *t*-test. ** *p* ≤ 0.01; *** *p* ≤ 0.001; **** *p* ≤ 0.0001. In all graphs, whenever the *p* value is not mentioned, differences between groups are not statistically significant (*p* > 0.05).

**Figure 2 ijms-25-01355-f002:**
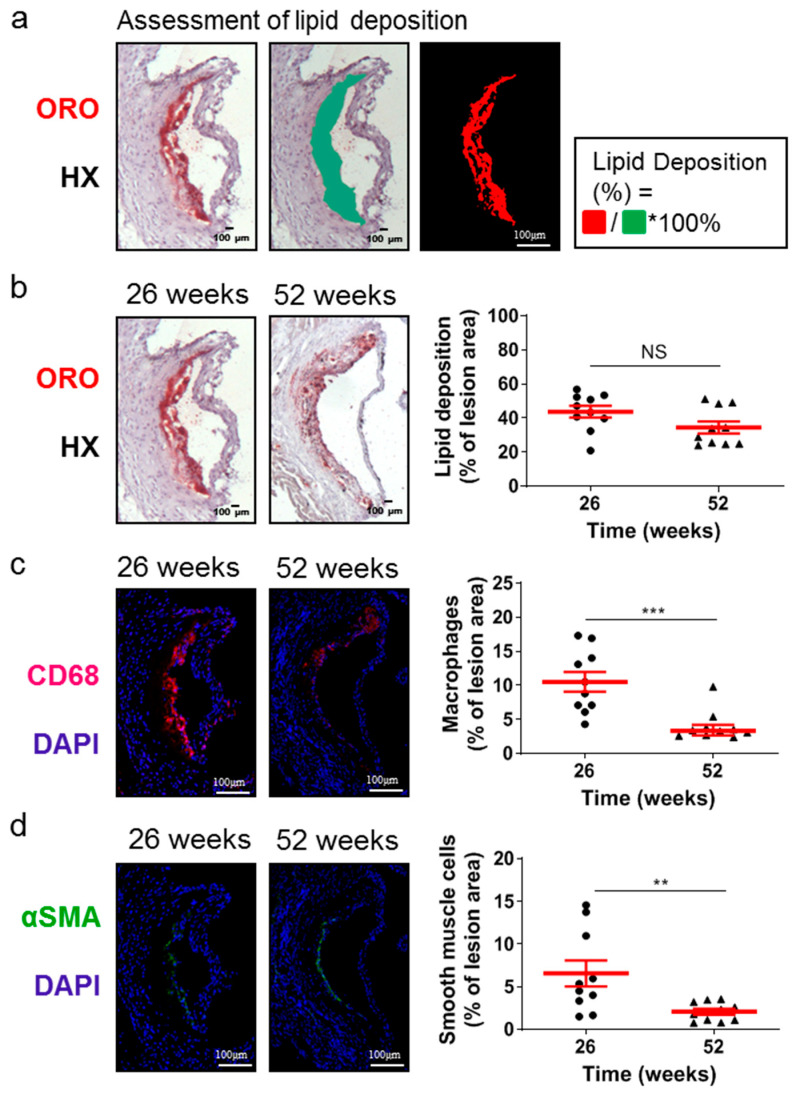
Morphological characteristics of atherosclerotic lesions of the aortic valve in 26- and 52-week-old *Apoe*-/- mice fed a normal chow diet. (**a**) Representative light photomicrographs for lipid deposition assessment. (**b**) Representative light photomicrographs and morphometric analysis of ORO-stained serial cross sections from the aortic root of 26- (circles) and 52-week-old (triangles) *Apoe*-/- mice. Results show the mean lipid-positive area ± SEM (red lines) of *n* = 10 mice per group, expressed as a percentage of the total lesion area. (**c**) Representative fluorescent photomicrographs and morphometric analysis of CD68-stained sections from the aortic root of 26- (circles) and 52-week-old (triangles) *Apoe*-/- mice. Results show the mean macrophage-positive area ± SEM (red lines) of *n* = 10 mice per group, expressed as a percentage of the total lesion area. (**d**) Representative fluorescent photomicrographs and morphometric analysis of alpha smooth muscle actin-stained sections from the aortic root of 26- (circles) and 52-week-old (triangles) *Apoe*-/- mice. Results show the mean smooth muscle cell-positive area ± SEM (red lines) of *n* = 10 mice per group, expressed as a percentage of the total lesion area. αSMA indicates alpha smooth muscle actin; DAPI, 4′,6-diamidino-2-phenylindole; HX, hematoxylin; ORO, Oil Red O. Statistical comparisons between groups were performed using the unpaired Student’s *t*-test. NS indicates no statistically significant differences between groups. ** *p* ≤ 0.01; *** *p* ≤ 0.001.

**Figure 3 ijms-25-01355-f003:**
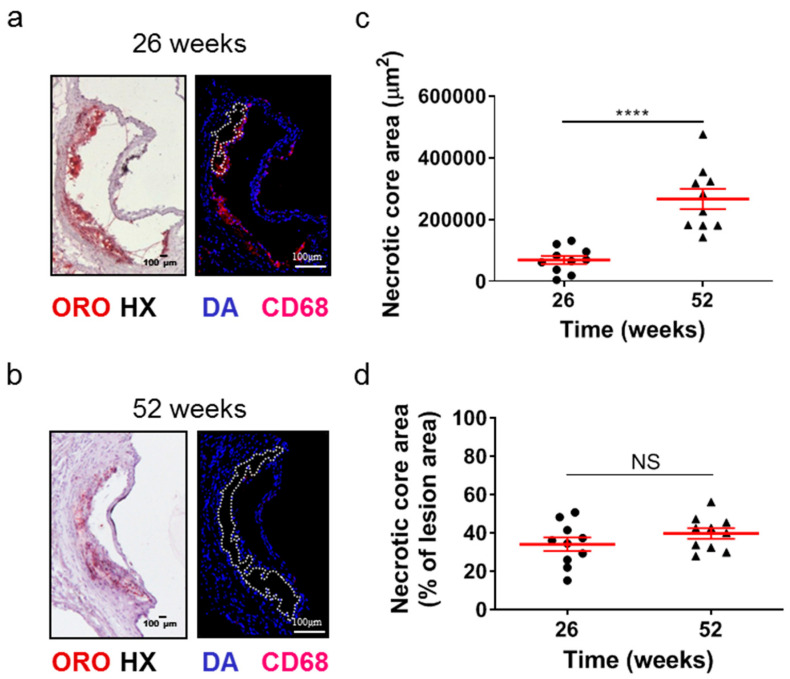
Morphometric analysis of the necrotic core area of the aortic roots of 26- and 52-week-old *Apoe*-/- mice fed a normal chow diet. (**a**,**b**) Representative light photomicrographs of ORO-stained cross sections and fluorescent photomicrographs of CD68/DAPI-stained sections (necrotic core areas outlined with dotted white lines) from the aortic root of 26- and 52-week-old *Apoe*-/- mice. (**c**,**d**) Morphometric analysis of the necrotic core area from CD68/DAPI-stained sections from the aortic root of 26- (circles) and 52-week-old (triangles) *Apoe*-/- mice. Results show the mean necrotic core area ± SEM (red lines) of *n* = 10 mice per group, expressed as an absolute value (**c**) or as a percentage of the total lesion area (**d**). Necrotic core area of atherosclerotic lesions was highly statistically significantly higher in 52- than in 26-week-old *Apoe*-/- mice (**c**, *p* < 0.0001); the percentage of necrotic core area out of lesion area followed a similar tendency, in spite of not reaching statistical significance (**d**, *p* = 0.225). DAPI indicates 4′,6-diamidino-2-phenylindole; HX indicates hematoxylin; and ORO indicates Oil Red O. Statistical comparisons between groups were performed using the unpaired Student’s *t*-test. NS indicates no statistically significant differences between groups. **** *p* ≤ 0.0001.

**Figure 4 ijms-25-01355-f004:**
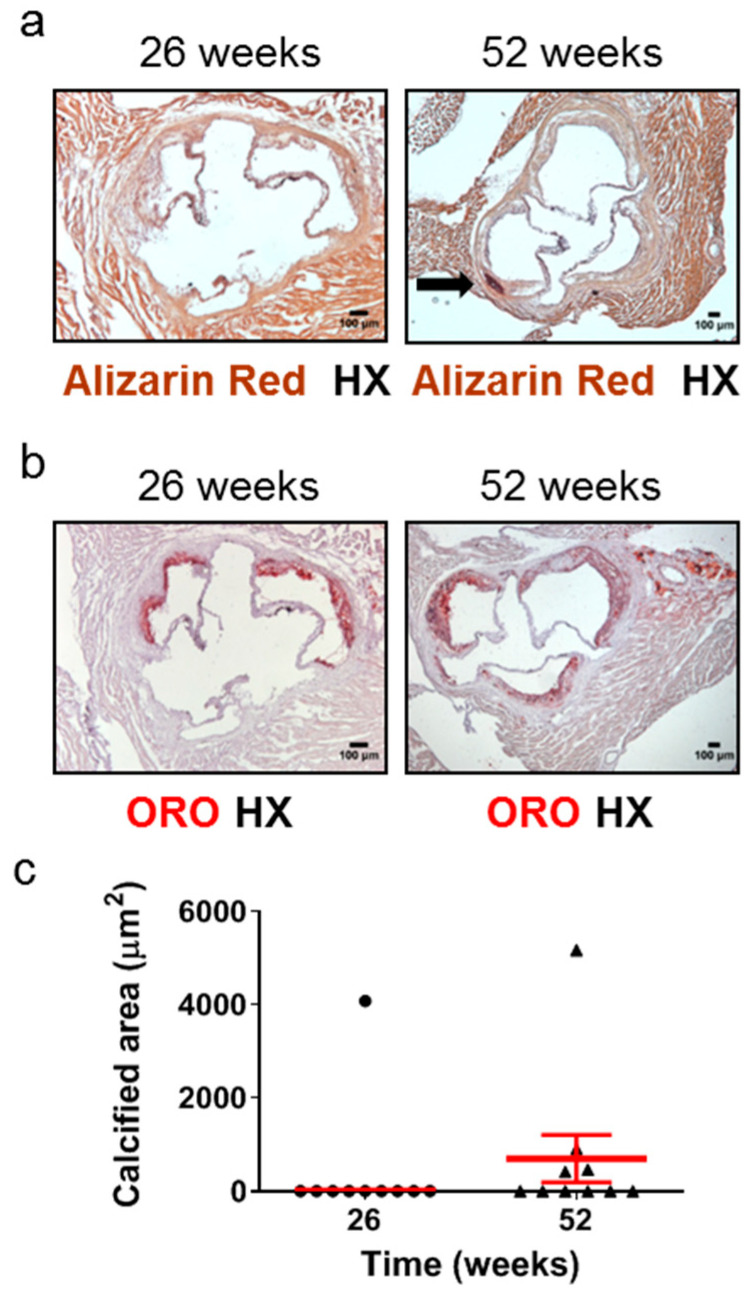
Development of calcification in atherosclerotic lesions of the aortic valve in 26- and 52-week-old *Apoe*-/- mice fed a normal chow diet. The progression of atherosclerosis towards late disease stages is coincident with a substantial increase in the frequency of the presence, but not necessarily also of the area, of aortic valve calcification. This histopathological characteristic of late disease stages is more rarely observed at intermediate stages. (**a**) Representative light photomicrographs of Alizarin Red S-stained (**a**) and ORO-stained (**b**) serial cross sections and morphometric analysis of the calcified area of atherosclerotic plaques on Alizarin Red S-stained serial cross sections (**c**) from the aortic root of 26- (circles) and 52-week-old (triangles) *Apoe*-/- mice. The black arrow depicts a calcium deposit (**a**). Results show the mean calcified area ± SEM (red lines) of *n* = 10 mice per group. It can be observed that while the area of aortic valve calcification is not necessarily higher at 52 weeks of age than at 26 weeks of age, the frequency of the presence of aortic valve calcification is substantially higher at 52 weeks of age (40%; in four out of 10 mice) than at 26 weeks of age (10%; in one out of 10 mice) (**c**). HX indicates hematoxylin; ORO indicates Oil Red O.

**Figure 5 ijms-25-01355-f005:**
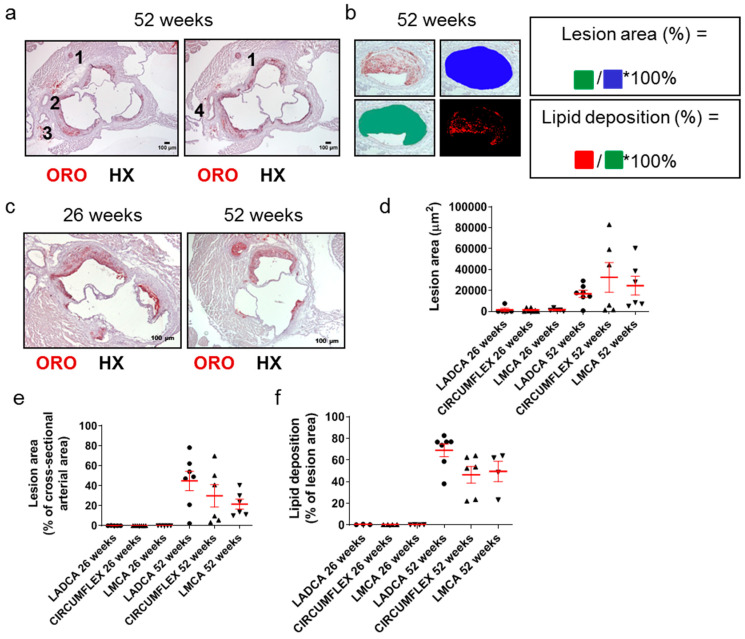
Morphological characteristics of atherosclerotic lesions in coronary arteries in 26- and 52-week-old *Apoe*-/- mice fed a normal chow diet. (**a**) Light photomicrographs of ORO-stained serial cross sections from the aortic root of a 52-week-old *Apoe*-/- mouse. Artery (1), which appears distant from arteries (2) and (3), is the right coronary artery. On the left image, artery (2) is the left anterior descending coronary artery (LADCA), since it appears more proximal to the right coronary artery (1) than artery (3); artery (3) is the circumflex coronary artery, as verified on the right image. The right image depicts the immediately subsequent serial section, in which the left anterior descending coronary artery (2) and the circumflex coronary artery (3) have converged at the level of the emergence of the left main coronary artery (LMCA) (4). HX indicates hematoxylin; ORO indicates Oil Red O. (**b**,**c**) Representative light photomicrograph of ORO-stained serial cross sections from the aortic root of a 26- and/or 52-week-old *Apoe*-/- mouse and schematics displaying the definitions of the percentage of lesion area out of the total cross-sectional arterial area, as well as the percentage of lipid deposition out of the total cross-sectional area occupied by an atherosclerotic plaque in a coronary artery, as follows. Lesion area (%) = cross-sectional area occupied by the atherosclerotic plaque/total cross-sectional area of the coronary artery * 100%. Lipid deposition (%) = cross-sectional area positively stained with ORO staining/cross-sectional area occupied by the atherosclerotic plaque * 100%. (**d**–**f**) Morphometric analysis of lesion area (**d**), the percentage of lesion area out of the total cross-sectional arterial area (**e**), and the percentage of lipid deposition out of the cross-sectional area occupied by the atherosclerotic plaque (**f**) in coronary arteries on ORO-stained serial cross sections from the aortic root of 26- and 52-week-old *Apoe*-/- mice. Results in each graph (**d**–**f**) show the mean ± SEM (red lines) of *n* = 5, 7, 5, 7, 6, 6 mice in the six groups, respectively. LADCA (circles); circumflex coronary artery (triangles); LMCA (triangles). It can be observed that lesion formation was sparse and restricted at 26 weeks of age but became highly frequent and extensive at 52 weeks of age.

**Table 1 ijms-25-01355-t001:** Ultrasonographic evaluation of 26-week-old and 52-week-old *Apoe*-/- mice fed a normal chow diet.

Parameter	*Apoe-/-* Mice 26 Weeks (*n* = 21)	*Apoe-/-* Mice 52 Weeks (*n* = 34)	*p* Value
Heart rate, beats/min	536.28 ± 6.41	568.61 ± 7.23	0.003 (**)
EDD, mm	3.04 ± 0.045	3.01 ± 0.05	0.695 (NS)
ESD, mm	1.60 ± 0.03	1.63 ± 0.03	0.465 (NS)
FS, %	47.05 ± 0.91	45.46 ± 0.66	0.157 (NS)
PWT, mm	0.71 ± 0.01	0.75 ± 0.01	0.009 (**)
r/h	2.12 ± 0.03	2.01 ± 0.04	0.063(NS)
Peak aortic velocity, cm/s	76.15 ± 2.40	86.09 ± 3.78	0.649 (NS)
Mean aortic velocity, cm/s	38.26 ± 1.50	43.78 ± 2.07	0.067(NS)
Peak aortic acceleration, m/s^2^	104.93 ± 2.55	105.77 ± 2.07	0.809 (NS)
Peak carotid velocity, cm/s	70.23 ± 3.06	74.05 ± 2.22	0.308 (NS)
Mean carotid velocity, cm/s	27.81 ± 1.33	31.67 ± 1.05	0.027 (*)
Carotid pulsatility index	2.69 ± 0.13	2.27 ± 0.11	0.019 (*)

Notes and abbreviations: Values of parameters are mean ± SD. EDD, end-diastolic diameter; ESD, end-systolic diameter; FS, fractional shortening; PWT, posterior wall thickness; r/h, ratio of left ventricular radius to PWT. Statistical significance values (*p*) are indicated with respect to inter-group comparisons using the unpaired Student’s *t*-test. NS indicates no statistically significant difference between the two groups. * *p* ≤ 0.05; ** *p* ≤ 0.01.

## Data Availability

The raw data supporting the conclusions of this article will be made available by the authors, without undue reservation.

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
