# Peer review of "Comprehensive Analysis of 1-Year-Old Female Apolipoprotein E-Deficient Mice Reveals Advanced Atherosclerosis with Vulnerable Plaque Characteristics"

_ijms, 2024, doi:10.3390/ijms25021355_

Round 1

Reviewer 1 Report

Comments and Suggestions for Authors

In the current manuscript, Kotsovilis et al thoroughly investigated the progression of the atherosclerotic plaque in the aortic valve regarding its composition and its effect on the cardiac function. Overall, this is a simple straightforward study, and indeed provide some interesting information, but there still some issues need to be addressed:

1.     Is there a specific reason that the authors only focus on female mice rather than both sexes?

2.     For panels in Figure 1, one way ANOVA should be used instead of Student’s t-test

3.     For Figure 1e, the author in the text that the quantitation is based on lesion area/ lesion size, which is not very clear how it is quantitated.

4.     Please show the quantitation of Figure 2b, though it doesn’t reach statistical significance.

5.     Does Figure 4B show the same thing as Figure 2B? It seems that both showed ORO staining of plaque?

6.     It is interesting that the author showed natural plaque formation in the coronary artery in Apoe knockout mice. Usually people check aorta, aortic root and carotid artery, where show the most severe plaque formation in mice. Did author check these region?

7.     Please also check the fibrosis cap and ECM content.

Reviewer 2 Report

Comments and Suggestions for Authors

The study by Kotsoviliset al. analyzes the atherosclerotic burden of female apoE-/- mice of 1 year of age. The study is technically sound. The data provided are of interest to the field, as this model is amply used. Further, using a standard chow diet, which has a slower effect than high-fat diets, adds value to the study, as it could be argued that it resembles human pathology more closely. 

A limitation of the study is the lack of an ApoE+/+ control group analyzed in the same conditions. However, considering that no plaque is expected to develop in WT animals, this is a minor limitation.

The authors should consider including "female" in the title, as this is an important feature of the study.

The authors could include the quantification of Figure 2b, 5a-c results. Figure 5 f is missing.

In the figures, sometimes the authors use Oil Red O and others the acronym ORO. The same expression could be used.

Round 2

Reviewer 1 Report

Comments and Suggestions for Authors

All my questions have been addressed, though it may be better to include a short paragraph of discussion to compare results measured via aortic root, whole aorta, carotid artery and coronary artery.